# Hydrogen Sulfide Promotes Adventitious Root Development in Cucumber under Salt Stress by Enhancing Antioxidant Ability

**DOI:** 10.3390/plants11070935

**Published:** 2022-03-30

**Authors:** Yayu Liu, Lijuan Wei, Li Feng, Meiling Zhang, Dongliang Hu, Jianzhong Tie, Weibiao Liao

**Affiliations:** 1College of Horticulture, Gansu Agricultural University, 1 Yinmen Village, Anning District, Lanzhou 730070, China; liuyayu199809@163.com (Y.L.); wlj920229@163.com (L.W.); feng1654114573@163.com (L.F.); h1468009331@163.com (D.H.); tiejianzhong1999@163.com (J.T.); 2College of Science, Gansu Agricultural University, Lanzhou 730070, China; zhangml@gsau.edu.cn

**Keywords:** hydrogen sulfide, osmotic substances, antioxidant system, lipid peroxidation, adventitious root development

## Abstract

As a gas signal molecule, hydrogen sulfide (H_2_S) can enhance plant stress resistance. Here, cucumber (*Cucumis sativus* ‘Xinchun NO. 4’) explants were used to investigate the role of H_2_S in adventitious root development under salt stress. The results show that sodium chloride (NaCl) at 10 mM produced moderate salt stress. The 100 µM sodium hydrosulfide (NaHS) treatment, a H_2_S donor, increased root number and root length by 38.37% and 66.75%, respectively, indicating that H_2_S effectively promoted the occurrence of adventitious roots in cucumber explants under salt stress. The results show that under salt stress, NaHS treatment reduced free proline content and increased the soluble sugar and soluble protein content during rooting. Meanwhile, NaHS treatment enhanced the activities of antioxidant enzymes [peroxidase (POD), superoxide dismutase (SOD), ascorbate peroxidase (APX) and catalase (CAT)], increased the content of ascorbic (ASA) and glutathione (GSH), reduced the content of hydrogen peroxide (H_2_O_2_) and the rate of superoxide radical (O^2−^) production, and decreased relative electrical conductivity (REC) and the content of malondialdehyde (MDA). However, the NaHS scavenger hypotaurine (HT) reversed the above effects of NaHS under salt stress. In summary, H_2_S promoted adventitious root development under salt stress through regulating osmotic substance content and enhancing antioxidant ability in explants.

## 1. Introduction

Adventitious rooting, a root formed by non-root tissues, is an important part of vegetative propagation. Adventitious roots not only have the functions of fixing and supporting plants, but also increase the ability of plants to absorb nutrients and water [1]. Both abiotic stress and biotic stress can make plant organs and tissues develop adventitious roots [2]. The formation of adventitious roots can expand plant root systems and give plants and cells the ability of regeneration. Study has shown that traditional plant hormones and small gaseous molecules, as signal molecules, participate in adventitious rooting [3]. Therefore, in-depth study of adventitious roots will help us to clarify the mechanism of plant growth and development.

During crop growth and development, biotic and abiotic stresses have an increasing impact on the yield and quality of agricultural products. Salt stress, as a major abiotic stress, severely affects plant growth and development. About 50% of irrigated land in the world is affected by salinization, and non-irrigated land in the region also undergoes salinization [4]. Excessive salinization of the soil seriously reduces the absorption of water and nutrients in plants, which in turn affects the normal metabolism in plants [5]. In addition, excessive intake of sodium (Na^+^) and chloride ion (Cl^−^) by plants may cause many adverse effects on plants, such as nutrient imbalance, cell membrane damage, and antioxidant system damage, even leading to plant death, and ultimately reducing crop yield and quality [6,7]. During growth and development, the water and mineral ions required by plants need to be absorbed from the rhizosphere environment through plant roots. Under salt stress conditions, the structure of the plant root system is affected. Salt stress may inhibit plant root cell division and cell elongation, and affect root tip meristem, thereby affecting the growth and development of plant roots [8]. Hydrogen sulfide (H_2_S) is a colorless, flammable gas with the smell of rotten eggs, and has long been considered a harmful gas. In 1996, H_2_S was proved to exist in the human body as a neuroactive substance for the first time, and people began to pay attention to its physiological functions [9]. Nowadays, H_2_S is considered to be the third gas signaling molecule after nitric oxide (NO) and carbon monoxide (CO), and it plays an essential role both in animals and plants. For example, H_2_S could promote seed germination [10] and root development, regulate stomatal movement [11], and keep flowers fresh [12]. Additionally, H_2_S is involved in plant responses to abiotic stresses, including drought stress [13], osmotic stress [14], salt stress [15], chilling stress [16], and heavy metal stress [17].

In recent years, there have been various studies on the roles of gas signal molecules in adventitious root generation under abiotic stress conditions. For example, H_2_ was associated with the development of adventitious root of cucumber under cadmium stress [17,18]. CO is associated with H_2_-induced adventitious root development in cucumber under stress of drought [19]. Liao et al. [20] found that NO and H_2_O_2_ could promoted adventitious root development in Marigold explants under drought stress. Nevertheless, there is little research on the roles of H_2_S in adventitious root formation under abiotic stresses. Hence, we speculate that H_2_S might promote adventitious root development under abiotic stress conditions. Therefore, in this study, sodium hydrosulfide (NaHS) was used to investigate the effect of H_2_S on adventitious root development in response to salt stress, and its specific mechanism was also revealed.

## 2. Results

### 2.1. Effects of Different Concentrations of NaCl on Adventitious Root Development

As shown in Figure 1, as the concentration of NaCl increases, both root number and length were decreased. The number of roots and root length in the 10 mM NaCl treatment were significantly lower than those in control and 8 mM NaCl treatments. Meanwhile, 10 mM NaCl treatments resulted in higher root number and length than 12, 14, and 16 mM NaCl treatments. Moreover, there was no significant change in root number and root length among 12, 14, and 16 mM NaCl treatments. These results indicate that treatments with 8, 10, and 12–16 mM NaCl could be termed as mild, moderate, and severe NaCl stress, respectively. As 10 mM NaCl induced moderate stress, the following experiments used the concentration.

### 2.2. Effects of Different Concentrations of NaHS on Adventitious Rooting under Salt Stress

To investigate the positive effect of H_2_S on adventitious rooting under salt stress, we treated an explant with NaHS of different concentrations under salt stress. NaCl treatment significantly reduced the number and length of roots in a concentration-dependent way (Figure 2A,B). Compared with NaCl treatment, treatments with 25, 50, 100, and 150 µM NaHS significantly increased root number by 62.25%, 82.42%, 122.48%, and 95.39%, respectively (Figure 2A,C). Additionally, compared to NaCl treatment, 25, 50, 100, and 150 µM NaHS treatments significantly increased root length by 44.51%, 35.10%, 78.08%, and 24.44%, respectively (Figure 2B,C). Among all concentrations of NaHS, 100 µM NaHS treatment produced the highest root number and the longest root length under salt stress, which has no significant difference with the control (Figure 2). Therefore, 100 µM NaHS was used for the following experiments.

### 2.3. Effects of H_2_S Scavenger HT on Adventitious Root Development under Salt Stress

To further confirm the role of H_2_S in adventitious root development under salt stress, the H_2_S scavenger HT was used in our study. Compared with NaCl treatment alone, NaCl + NaHS treatment significantly increased root number and root length by 38.37% and 66.7%, respectively (Figure 3A,B). Additionally, in comparison to NaCl + NaHS treatment, NaCl + NaHS + HT treatment significantly decreased root number and root length by 18.43% and 27.30%, which further indicates the key roles of H_2_S in inducing the adventitious root development when facing salt stress. Similarly, when the explants were grown in substrate for 20 days, NaHS + NaCl treatment significantly enhanced the growth of cucumbers in comparison with NaCl treatment alone (Figure 3D). Thus, these results suggest that the addition of NaHS significantly reduced the damaging effect of salt stress on cucumber seedlings, inducing adventitious root development and then promoting cucumber growth.

### 2.4. Effects of H_2_S on the Content of Osmotic Substances during Adventitious Rooting under Salt Stress

In comparison with the control, NaCl treatment significantly decreased the content of soluble sugar and soluble protein (Figure 4A,B), showing that salt stress induces osmotic stress in the explants. In addition, the content of soluble sugar and soluble protein was significantly increased after adding NaHS under salt stress. Additionally, compared with NaCl + NaHS treatment, NaCl + NaHS + HT treatment significantly reduced soluble sugar and soluble protein. On the contrary, the proline content was increased by NaCl treatment (Figure 4C). Compared with NaCl treatment alone, NaCl + NaHS treatment decreased proline content, which was reversed in NaCl + NaHS + HT treatment. These results mentioned above suggest that H_2_S could regulate the levels of osmotic substances during adventitious rooting under salt stress.

### 2.5. Effect of H_2_S on the Degree of Membrane Lipid Peroxidation during Adventitious Root Development under Salt Stress

To investigate the effect of NaHS on the plasma membrane of explant under salt stress, the relative conductivity and MDA content were measured in this experiment. Compared with the control, NaCl treatment significantly increased REC and MDA content by 62.46% and 102.11%, respectively (Figure 5). In contrast to NaCl treatment alone, NaHS + NaCl treatment led to a marked decline in the REC and MDA content. However, the REC and MDA content in NaCl + NaHS + HT treatment were significantly higher than those in NaCl + NaHS treatment. These results indicate that NaHS significantly reduced salt stress-induced damage to cell membranes during adventitious rooting.

### 2.6. Effect of H_2_S on the O^2−^ Production Rate and H_2_O_2_ Content during Adventitious Root Development under Salt Stress

As shown in Figure 6, compared with the control, salt stress treatment significantly increased the O^2−^ production rate and H_2_O_2_ content of cucumber explants, further exacerbating the degree of membrane lipid peroxidation. However, this increase was inhibited in the presence of NaHS. NaHS + NaCl treatment led to a marked decrease in O^2−^ production rate and H_2_O_2_ content by 44.73% and 36.24%, respectively, as compared to the NaCl treatment alone. Compared to the NaCl + NaHS treatment, NaCl + NaHS + HT treatment significantly increased the O^2−^ production rate and H_2_O_2_ content by 45.77% and 29.41%, respectively, implying the roles of H_2_S in decreasing membrane lipid peroxidation. Therefore, these results suggest that the application of exogenous NaHS may inhibit the accumulation of ROS and thus protect cucumber plants from oxidative stress.

### 2.7. Effect of H_2_S on Antioxidation Abilities during Adventitious Root Development under Salt Stress

As is shown in Figure 7A–D, the activities of all four enzymes—POD, SOD, APX, and CAT—were significantly increased under salt stress. Meanwhile, compared with NaCl treatment alone, the addition of exogenous NaHS significantly increased the activities of POD, SOD, APX, and CAT. However, the addition of HT significantly decreased the activities of these four enzymes compared to NaCl + NaHS treatment. Thus, NaHS increased the activities of antioxidant enzymes during adventitious root development under salt stress. As shown in Figure 7E,F, ASA and GSH contents were significantly reduced under NaCl treatment compared with the control. However, compared to NaCl treatment, when NaHS was applied, ASA and GSH contents were increased by 49.14% and 43.50%, respectively. After the addition of HT, both ASA and GSH contents decreased to different degrees than the NaCl + NaHS treatment, with no significant difference in ASA content and a significant decrease of 45.28% in GSH content. These results imply that NaHS application enhances the antioxidant capacity of explants in response to NaCl stress.

## 3. Discussion

Salt stress is an essential limiting element in plant growth and development [21]. The generation of adventitious roots is a reflection of plant propagation ability. The formation of adventitious roots expands the root system of plants and increases plant tolerance under stress conditions [22]. In this study, salt stress reduced the number and length of adventitious roots in cucumber (Figure 1). Previous study has shown that H_2_S might play a critical role in response to abiotic stress in plants [23]. H_2_S also may regulate many aspects of plant nutritional and reproductive growth, such as improved seed germination rate and adventitious root induction [24]. In our preliminary experiment, both morpholin-4-ium 4-methoxyphenyl(morpholino)phosphinodithioate (GYY 4137) and NaHS were used to screen the most suitable H_2_S concentration. The experimental results showed that both GYY 4137 and NaHS could promote adventitious rooting, but the positive roles of GYY 4137 under salt stress was much less than that those of NaHS. Thus, NaHS was chosen as the H_2_S donor in the experiment. Our results demonstrate that NaHS-induced adventitious root formation in cucumbers under salt stress in a concentration-dependent manner, and 100 μM NaHS obtained the maximum biological effect (Figure 2). Our results are consistent with the results of Deng et al. [25], who reported that H_2_S could alleviate the inhibition of salt stress on the growth of wheat seedlings. Meanwhile, Zhang et al. [24] found that NaHS increased root number and length in excised willow (*Salix matsudana* var. *tortuosa* Vilm) and soybean (*Glycine max* L.) seedlings. Moreover, our results suggested that H_2_S could strengthen the antioxidant system, reduce lipid peroxidation and the accumulation of reactive oxygen species, and relieve oxidative stress caused by salt, which is consistent with the results from Jiang et al. [26]. Overall, these results suggest that H_2_S reduces root injury from salt stress by increasing adventitious root development.

Plants can alleviate the adverse effects of salt stress by accumulating soluble proteins, amino acids, soluble sugars, and other small molecular organic substances [27]. Soluble sugars can maintain intracellular osmotic pressure [28]. In addition to participating in osmotic regulation, soluble proteins also partly represent the change of plant organ function [29]. The increase of proline content can also regulate plant osmotic potential and improve plant tolerance [30]. In our study, salt stress significantly reduced the contents of soluble sugar and soluble protein in cucumber explants, while it increased the proline content (Figure 4). It indicates that the breakdown of soluble proteins may produce many free amino acids that facilitate osmoregulation by the root system. Therefore, the decomposition of soluble protein under salt stress may be one of the reasons for the increase of proline content. Meanwhile, NaHS treatment significantly reduced the proline content but increased the soluble sugar and soluble protein level, indicating that H_2_S might maintain osmotic pressure and enhance rooting under salt stress. While, a NaHS scavenger HT decreased the soluble sugar and soluble protein content and increased the proline content (Figure 4). The results seem to be consistent with those of Wei et al. [12], which show that NaHS application increases the content of soluble sugar and soluble protein in cut rose and chrysanthemum flowers. Meanwhile, Li et al. [31] found that NaHS application also increased the content of soluble sugar and soluble protein in Lanzhou lily. It was also found that the exogenous H_2_S application enhanced soluble sugar content and decreased proline content in zucchini under nickel stress [32]. Therefore, H_2_S may promote the emergence of adventitious roots under salt stress by maintaining and adjusting osmotic pressure.

Under salinity stress, plants usually produce excess reactive oxygen species (ROS) such as O^2−^ and H_2_O_2_ [33]. Excessive accumulation of ROS results in impaired membrane integrity and oxidative damage in plants. Plants enhance antioxidant capacity by removing accumulated reactive oxygen species [34]. REC and MDA content can usually reflect the status of plant cell membrane and lipid peroxidation [35]. Here, NaCl treatment significantly increased the O^2−^ production rate, H_2_O_2_ content, and REC and MDA content in cucumber explants (Figure 5 and Figure 6), suggesting that salt stress may lead to the imbalance of ROS metabolism, cause membrane lipid peroxidation, and increase cell membrane permeability. However, H_2_S significantly decreased the O^2−^ production rate, H_2_O_2_ content, and REC and MDA content, which effectively alleviated the inhibition caused by salt stress to adventitious rooting. Similar results were reported in *Spinacia oleracea* L. seedlings, where they found that NaHS decreased ROS accumulation and reduced MDA content under drought stress [36]. Meanwhile, Ahmad et al. [37] found that NaHS increased electrolyte leakage (EL) and H_2_O_2_ and MDA content in cauliflower under Cr stress. Moreover, Ding et al. [38] found that exogenous H_2_S application reduced H_2_O_2_ and MDA content in wheat seedlings under salt stress. Thus, H_2_S may enhance adventitious root formation by reducing membrane lipid peroxidation and maintaining cell membrane integrity under salt stress.

The antioxidant system developed during plant evolution contributes to the balance of ROS metabolism and directly reflects plant salt tolerance [39]. For instance, among the antioxidant enzymes, SOD, CAT, GPX, APX, and GR are vital enzymes to scavenge intracellular ROS [40]. The increased antioxidant defenses are positively correlated with the reduced oxidative damage in plants under abiotic stress [41]. In this study, salt stress significantly increased the activities of POD, SOD, APX, and CAT during rooting (Figure 7). Guo et al. [42] found that the antioxidant enzyme activities in wheat were significantly increased under cadmium stress. Our results show that NaHS treatment could also significantly increase the activities of POD, SOD, APX, and CAT under salt stress (Figure 7), confirming that H_2_S scavenges O^2−^ content in the explants by increasing the activity of antioxidant enzymes, thus reducing the damage of salt stress on adventitious root development. The addition of NaHS was also found to significantly increase antioxidant enzyme activity in wheat and tomato [38,43]. Moreover, addition appropriate doses of NaHS can effectively activate antioxidant enzyme activity and prolong the postharvest freshness in cut flowers [12]. Therefore, the increase in antioxidant enzyme activity reduced ROS accumulation, thereby alleviating salt stress damage on adventitious root formation and further improving salt tolerance during rooting.

ASA and GSH are two vital antioxidants which can assist antioxidant enzymes in H_2_O_2_ metabolism, thus scavenging ROS overproduction [44]. In this study, exogenous H_2_S application led to a significant increase in ASA and GSH contents during adventitious rooting under NaCl stress (Figure 7). Our results indicate that H_2_S could maintain cellular oxidative homeostasis by increasing antioxidant contents, thereby mitigating cytotoxicity and oxidative damage caused by ROS accumulation, and thus promoting rooting.

## 4. Materials and Methods

### 4.1. Plant Material and Growth Conditions

*Cucumis sativus* ‘Xinchun 4’ seeds (Gansu Academy of Agricultural Sciences, Lanzhou, China) were soaked in room temperature water for 5 h. After that the seeds were fished out and placed in a warm environment to germinate. The main roots were removed after 5 days of growth. The explants were placed in conical flasks containing distilled water or the compounds, maintained at a temperature of 25 °C, set with appropriate light intensity and photoperiod. After 5 days of growth again, the number and length of adventitious roots of each explant were counted and recorded, and photographed. In addition, the explants were planted in substrate and managed normally, and the overall morphology of plants was photographed and recorded after 20 days.

### 4.2. Explant Treatments

Cucumber explants were cultured in an artificial intelligence type of light incubator (HGZ-H400, Shanghai Xinnuo Instrument Group Co. LTD, Shanghai, China) for 5 days with 50 mL distilled water (the control) or various concentrations of NaCl solution (8, 10, 12, 14, and 16 mM). The number and length of the roots were measured after the treatment, and the appropriate NaCl concentration was evaluated.

After selecting the appropriate NaCl concentration, the H_2_S donor NaHS concentration was screened. In this experiment, a total of 4 NaHS gradients were set up: 0 (distilled water, the control), NaCl, 25 μM NaHS + NaCl, 50 μM NaHS + NaCl, 100 μM NaHS + NaCl, 150 μM NaHS + NaCl. The cucumber explants were incubated with the above six solutions in an artificially intelligent lighted incubator for 5 days. The experiment was repeated three times, 10 seedlings per replicate, and a total of 180 seedlings.

After the appropriate NaHS concentration was selected, the cucumber explants were treated with the H_2_S scavenger (HT). In this experiment, four different treatments were used: distilled water (the control), NaCl, NaCl + NaHS and NaCl + NaHS + HT. The experiment was repeated three times, 10 seedlings per replicate, and a total of 180 seedlings.

### 4.3. Determination of Soluble Sugar, Soluble Protein Content

Soluble sugar content was measured using the method of Van Handel [45]. About 0.2 g of cucumber explants were ground and mixed with 10 mL of distilled water in a test tube. The samples were boiled in a water bath for 30 min and the supernatant was collected. This step needs to be repeated twice. Next, 0.5 mL supernatant and 7 mL of reaction solution [1.5 mL of distilled water, 0.5 mL of ethyl anthranilate, and 5 mL of 98% H_2_SO_4_] were mixed in a boiling water bath for 1 min and then cooled to room temperature. The absorbance was monitored at 630 nm.

Soluble protein content was determined using the Kučerová [46] method. A total of 0.2 g of sample was added to 5 mL of distilled water, was ground into a homogenate, and was centrifuged at 4 °C, 12,000× *g* for 20 min to collect the supernatant. A total of 1 mL of the supernatant was transferred to a test tube and 5 mL of Coomassie blue G-250 solution was added. After standing for 2 min, colorimetric and absorbance were measured at 595 nm. The protein content was calculated by the standard curve.

Proline content was measured using the Storey [47] method. The 0.2 g cucumber explants were cut into small pieces, put in a test tube, and 5 mL 3% sulfosalicylic acid was added to obtain the proline extract in a boiling water bath for 15 min. The 2 mL of glacial acetic acid and acid ninhydrin were added to the extract, which was then heated in a boiling water bath for 30 min. Next, extraction solution was cooled to room temperature and 5 mL of toluene was added. The mixture was shaken to extract the red product and left in the dark for 20 min. The absorbance value at 520 nm was measured and recorded.

### 4.4. Determination of REC and MDA Content

Relative electrical conductivity (REC): 0.1 g of cucumber explants were placed in 10 mL of deionized water in a test tube and soaked in a thermostat at 25 °C for 2 h to measure the electrical conductivity R1. After this, the sample is boiled in a water bath for 30 min, and once the temperature has dropped to room temperature, the conductivity R2 can be measured. The content of REC can be calculated by the formula which is R1/R2 × 100%.

Malondialdehyde (MDA): 0.2 g of cucumber explants were placed in pre-chilled mortar, 5 mL of TCA was added and ground, the homogenate was transferred to a test tube. After centrifugation at 4 °C, 12,000× *g* for 15 min, the supernatant was aspirated. A total of 1 mL of supernatant was mixed with 0.5% TBA solution. The mixture was boiled in a water bath for 30 min, rapidly cooled, and centrifuged for 10 min. The supernatant was taken to determine the absorbance at 450 nm, 532 nm, and 600 nm.

### 4.5. Determination of the O^2−^ Production Rate and H_2_O_2_ Content

A 0.5 g quantity of sample was ground in 1 mL of PBS buffer (pH 7.8) and centrifuged for 15 min (4 °C, 12,000× *g*). Totals of 1 mL of PBS buffer and 1 mL of hydroxylamine chloride reagent were added to 1 mL of the supernatant mix and left at room temperature for 1 h. A total of 0.5 mL of the incubation solution was mixed with 1 mL of 17 mM *p*-aminobenzene sulfonic acid and 1 mL of 7 mM α-naphthylamine. This reaction mixture was left at 25 °C for 20 min and the absorbance was read at 530 nm.

A 0.5 g quantity of sample was ground with pre-chilled acetone. The homogenate was centrifuged at a low temperature for 20 min, and 1 mL of supernatant was mixed with 0.1 mL of 10% TiCl_4_ and 0.2 mL of NH_3_·H_2_O. After reacting for 5 min, this was further centrifuged at 12,000× *g* and a temperature of 4 °C for 15 min. The precipitate was collected, 3 mL of 2 M H_2_SO_4_ was added, and the absorbance was measured at 415 nm.

### 4.6. Determination of Antioxidant Enzymes and Antioxidants

To extract the enzyme solution, three cucumber explant samples were collected randomly from each treatment. A 0.5 g quantity of sample was ground to powder with liquid nitrogen and extracted with 5 mL of 0.05 M phosphate buffer (pH 7.8). The homogenate was centrifuged for 15 min (12,000× *g*, 4 °C) and the supernatant was used for subsequent enzyme activity assays.

Peroxidase (POD) activity: the supernatant (100 μL) was added to 2.6 mL of 0.3% guaiacol and 0.3 mL of 0.6% H_2_O_2_, and the reaction was stopped by adding 2 mL of 20% TCA in an ice bath. Next, the OD value was measured at 470 nm.

Superoxide dismutase (SOD) activity: the reaction system included 3 mL of reaction mixture [1.5 mL phosphate buffer (0.05 M, pH 7.8), 0.3 mL methionine (130 mM), 0.3 mL NBT solution (750 μM), 0.3 mL ethylene diamine tetraacetic acid (1 mM), 0.3 mL riboflavin (20 μM) and 0.3 mL distilled water] and 20 μL of enzyme solution. Quantities of 3 mL of reaction mixture and 20 μL of PBS buffer were added to the control tube and covered with tin foil to block the light. The reaction tube was reached at 4000 Lux light at 25 °C for 20 min. After the reaction was finished, the absorbance was monitored at 560 nm.

Ascorbate peroxidase (APX) activity: 100 μL of the supernatant was added to 2.6 mL of reaction solution [containing potassium phosphate buffer (pH 7.0), 0.5 mM ascorbic acid, and 2 mM H_2_O_2_)], and the OD was measured at 290 nm.

Catalase (CAT) activity: 2.9 mL of reaction solution [containing 50 mM potassium phosphate buffer (pH 7.0) and 20 mM H_2_O_2_] was mixed with 100 μL of supernatant, and the absorbance was monitored at 240 nm.

Ascorbic acid (AsA): samples (0.5 g) were grinded and centrifuged. The mixture of 0.2 mL of supernatant, 0.5 mL of PBS buffer (pH 7.4), and 0.1 mL of 10 mM dithiothreitol (DTT) was placed in the reaction solution for 60 min at 40 °C. The absorbance was measured at 525 nm.

Glutathione (GSH): The reaction mixture contained 0.2 mL supernatant, 0.05 mL H_2_O, 0.5 mL of 2.5 mM EDTA, 0.1 mL of 0.5 mM NADPH, and 0.1 mL of 6 mM 2-nitrobenzoic acid. The absorbance was measured at 412 nm.

### 4.7. Statistical Analysis

Data were analyzed using SPSS 22.0 software (SPSS Inc., Chicago, IL, USA). All data were expressed as the mean ± standard error (SE) of three independent replicates. Duncan’s analytical test (*p* < 0.05) was used to determine the significance of the differences between treatments.

## 5. Conclusions

In summary, H_2_S significantly promoted the occurrence of adventitious roots in cucumber explants under salt stress. Through further studies, it was found that H_2_S could promote the accumulation of osmoregulatory substances (soluble sugar and soluble protein) and increase the activities of antioxidant enzymes (POD, SOD, APX, and CAT) and the content of antioxidants (ASA and GSH) during adventitious root formation under salt stress. Besides, H_2_S reduced proline and MDA content and REC during that process. Overall, our study provides evidence that H_2_S could reduce the damage of salt stress on adventitious root development by regulating the accumulation of osmoregulatory substances and antioxidants, the activity of antioxidant enzymes, as well as lipid peroxidation and cell membrane stability in cucumber. Thus, our findings may provide new insights into the mechanisms of H_2_S-induced adventitious rooting under salt stress. However, at present, there are few reports on the signaling mechanisms of plant H_2_S in response to abiotic stresses. Therefore, the deeper mechanisms of H_2_S-regulated rooting under salt stress conditions need to be further investigated, including the mining of critical genes and receptors.

## Figures and Tables

**Figure 1 plants-11-00935-f001:**
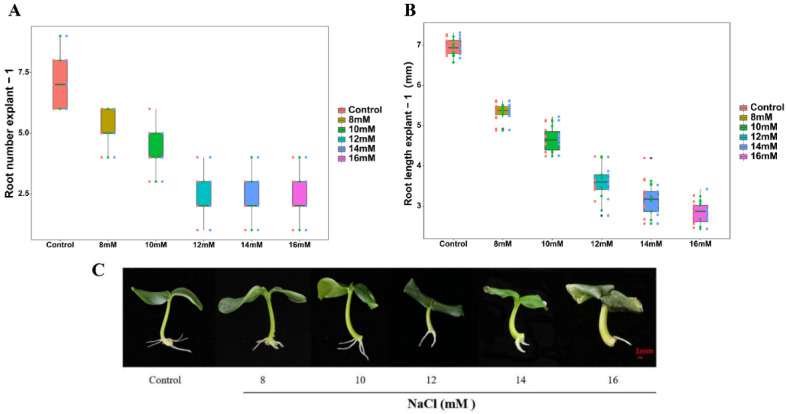
Effects of different concentrations of NaCl on adventitious root development in cucumber explants. The primary roots were removed from hypocotyl of 5-day-old seedlings. Explants were incubated for 5 days with distilled water (the control) and different concentrations of NaCl (0, 8, 10, 12, 14, and 16 mM). The numbers (**A**) and root length (**B**) of adventitious root were expressed as mean ± SE (*n* = 3, 10 explants were used per replicate). Values are means of three independent replicates. Photos [overall plant morphology (**C**)] of the plants were taken 5 days after treatment.

**Figure 2 plants-11-00935-f002:**
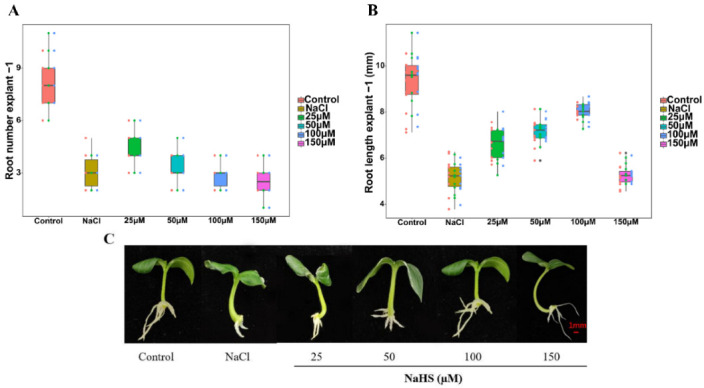
Effects of different concentrations of NaHS on adventitious root development in cucumber explants. The primary roots were removed from the hypocotyl of 5-day-old seedlings. Explants were incubated for 5 days with distilled water (the control) and different concentrations of NaHS (25, 50, 100, and 150 µM) + 10 mM NaCl. The number (**A**) and length (**B**) of adventitious roots were expressed as mean ± SE (*n* = 3, 10 explants were used per replicate). Values are means of three independent replicates. Photos [overall plant morphology (**C**)] of the plants were taken 5 days after treatment.

**Figure 3 plants-11-00935-f003:**
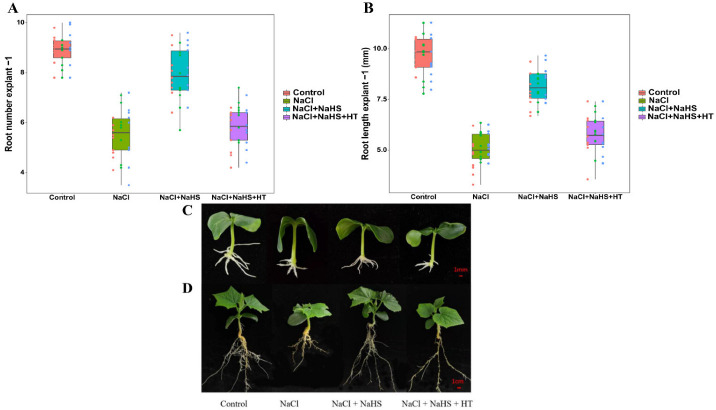
Effects of H_2_S scavenger HT on adventitious root development under salt stress. The primary roots were removed from hypocotyl of 5-day-old seedlings. Explants were incubated for 5 days with distilled water (the control), NaCl, NaCl + NaHS, NaCl + NaHS + HT. The number (**A**) and length (**B**) of adventitious roots were expressed as mean ± SE (*n* = 3, 10 explants were used per replicate). Values are means of three independent replicates. Photos [overall plant morphology of explants (**C**)] of the plants were taken 5 days after treatment. Photos [overall plant morphology (**D**)] of the plants were taken 20 days after treatment.

**Figure 4 plants-11-00935-f004:**
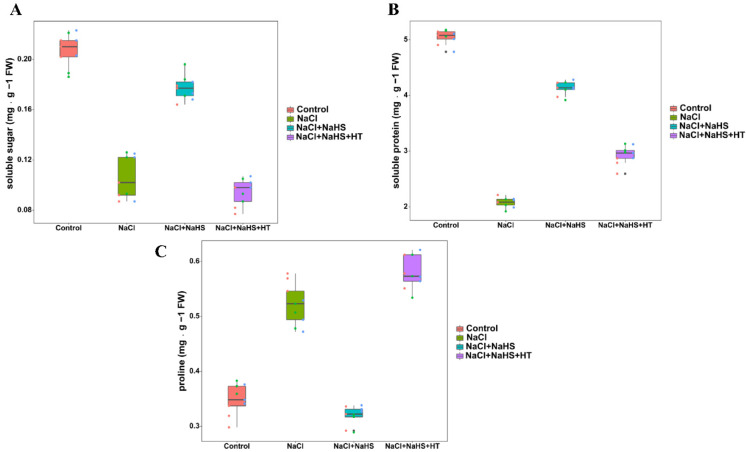
Effect of NaHS application on soluble sugar content (**A**), soluble protein content (**B**), and proline content (**C**) of explants species. The primary roots were removed from hypocotyl of 5-day-old seedlings. Explants were incubated for 5 days with distilled water (the control), NaCl, NaCl + NaHS, NaCl + NaHS + HT. Values are means of three independent replicates.

**Figure 5 plants-11-00935-f005:**
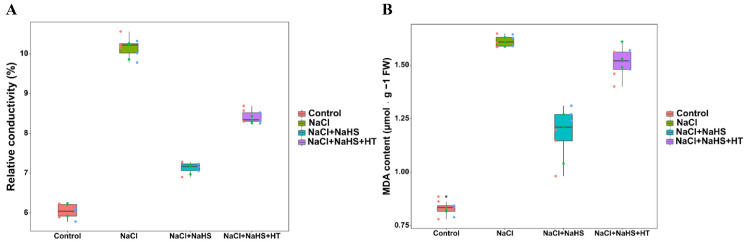
Effect of NaHS application on Relative conductivity (**A**) and MDA content (**B**) of explants species. The primary roots were removed from hypocotyl of 5-day-old seedlings. Explants were incubated for 5 days with distilled water (the control), NaCl, NaCl + NaHS, NaCl + NaHS + HT. Values are means of three independent replicates.

**Figure 6 plants-11-00935-f006:**
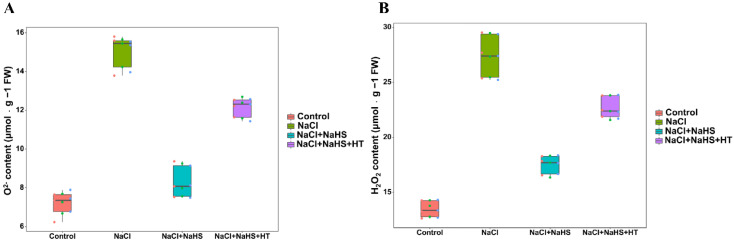
Effect of NaHS application on O^2−^ content (**A**) and H_2_O_2_ content (**B**) of explants species. The primary roots were removed from hypocotyl of 5-day-old seedlings. Explants were incubated for 5 days with distilled water (the control), NaCl, NaCl + NaHS, NaCl + NaHS + HT. Values are means of three independent replicates.

**Figure 7 plants-11-00935-f007:**
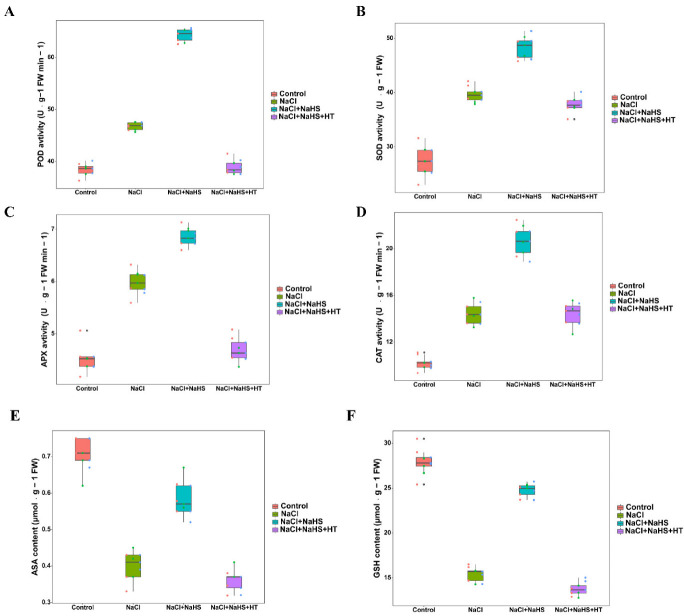
Effect of NaHS application on antioxidant enzyme activity [POD (**A**), SOD (**B**), APX (**C**), CAT (**D**)] and antioxidants [ASA (**E**) and GSH (**F**)] in explants. The primary roots were removed from hypocotyl of 5-day-old seedlings. Explants were incubated for 5 days with distilled water (the control), NaCl, NaCl + NaHS, NaCl + NaHS + HT. Values are means of three independent replicates.

## Data Availability

All data, tables and figures in this manuscript are original.

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
