# Peer review of "Hydrogen Sulfide Promotes Adventitious Root Development in Cucumber under Salt Stress by Enhancing Antioxidant Ability"

_plants, 2022, doi:10.3390/plants11070935_

Round 1

Reviewer 1 Report

The authors presents their interesting finding on the effect of hydrogen sulphide in alleviating salt stress in cucumber plants. The authors has also depicted how the use of an NaHS scavenger, hypotaurine reverses the alleviating effect of hydrogen sulphide donor NaHS. The authors depicted the effect of different combinations of treatment on the redox status of the plants.

But the authors has made these studies in the seedling stage of the plant. It would be very interesting to know about the responses of similar treatments in comparatively older plants and also in soil. I think depiction of plant responses at some advance stages will make the manuscript more interesting and relevant.

At least the authors can show morpho-physiological data at some advance stages with pictures and that will suffice.

Reviewer 2 Report

The work presented by Weibiao Liao and colleagues, entitled “Hydrogen Sulfide Promotes Adventitious Root Development in Cucumber under Salt Stress by Enhancing Antioxidant Ability” Explores the effect of the H2S donor NaHS on adventitious root growth in salt stressed cucumber explants. Although H2S was reported to participate in root growth processes in different species, the is a lack of information no the role of H2S on the Cucumber explants experimental systems. In the present work the authors reports that NaHS alleviates the deleterious effect of NaCl stress, by modulating the homeostasis of the osmoregulatory substances and REDOX homeostasis . Although the work is novel, and add knowledge to understanding the role of this gasotransmiter on root growth and development, there are several flaws that should be addressed to make it suitable for publications-

1.- Fig1C and Fig 2C. I can’t see a correlation between the length of the adventitious roots shown in the pictures and the plotted values. E.g. In Fig 2C doing a fast measurement with an image editor I got an average of aprox 14 mm for the explant of 150 uM treatment, which is about twice the size plotted in fig 2B. May be there is a mistake with the size of the scale bar, or with the election of the representative image. In any case, it'll be good to see all the data points in the chart as in a Bar plot + jitterplot, or a Box Plot + jitter plot

2.- Introduction: The is a format error in the MS and none of the citations can be read.

3.- Results: The text of the Results section is rather scarce, it is limited to the description of the figures

4.- Materials and Methods: Explants treatments: It is not clear, from this section, how and when NaHS was applied to the explants. If, as I presume, the NaHS treatments were performed for 5 days, as in the NaCl treatments, it’ll be interesting to see the effect of slow-rate realising donor as GYY 4137, to compare with the effect of the fast releasing donor NaHS.

Please do state how many explants per independent replicas were used?

5.- I would be useful to have more complete legends to figures, in order to have a clearer idea of how the data was obtained, without going to the M&M section.

Round 2

Reviewer 1 Report

I think the manuscript can now be accepted. Fig 3 has been modified by the authors during the revision process. The old image is still present in the manuscript. I think it has to be deleted

Reviewer 2 Report

The authors has thoroughly addressed most my concerns, and are now presented and improved version of the MS, although there is still a few issued that should be attended to make it suitable for publication in MDPI.

Figures:

The aim o the jitter plot is to see all the measurements in order to see, graphically, the distribution of the data points. If the data comes from 3 independent replica with 10 explants each, It would be good to see all the 30 data points in the plot. What is the mean of each of those 3 points? Are they averages?  In addition one can group, by color or symbol, the data points corresponding to a single replica.

There is no units on the axes of the new plots

Figs 1 and 3 show a scale bar in mm and Fig 2 a scale bar in cm. Is that so?

Fig 3 C & D: It does not make much sense that 5 days old and 20 days old share the same scale bar

Response point 2. Regarding your question about comparing the effect of GYY 4137 with NaHS on

adventitious root development, our laboratory has done relevant experiments before and the results were not satisfactory, so GYY 4137 was not used in the experiments.

Responde point 2. Regarding your question about comparing the effect of GYY 4137 with NaHS on

adventitious root development, our laboratory has done relevant experiments before and the results were not satisfactory, so GYY 4137 was not used in the experiments.

What does “not satisfactory mean” It has no effect? Is it possible to mention it in the discussion?
